# Differential Bias for Creatinine- and Cystatin C- Derived Estimated Glomerular Filtration Rate in Critical COVID-19

**DOI:** 10.3390/biomedicines10112708

**Published:** 2022-10-26

**Authors:** Anders O. Larsson, Michael Hultström, Robert Frithiof, Ulf Nyman, Miklos Lipcsey, Mats B. Eriksson

**Affiliations:** 1Department of Medical Sciences, Section of Clinical Chemistry, Uppsala University, 751 85 Uppsala, Sweden; 2Department of Surgical Sciences, Anaesthesiology and Intensive Care Medicine, Uppsala University, 751 85 Uppsala, Sweden; 3Department of Medical Cell Biology, Integrative Physiology, Uppsala University, 751 23 Uppsala, Sweden; 4Department of Epidemiology, McGill University, Montréal, QC H3A 0G4, Canada; 5Lady Davis Institute of Medical Research, Jewish General Hospital, Montréal, QC H3T 1E2, Canada; 6Department of Translational Medicine, Division of Medical Radiology, Lund University, 221 85 Malmö, Sweden; 7Hedenstierna Laboratory, Department of Surgical Sciences, Uppsala University, 751 85 Uppsala, Sweden; 8NOVA Medical School, New University of Lisbon, 1099-085 Lisbon, Portugal

**Keywords:** biomarkers, corticosteroids, creatinine, COVID-19, cystatin C, eGFR, intensive care, kidney, SARS-CoV-2

## Abstract

COVID-19 is a systemic disease, frequently affecting kidney function. Dexamethasone is standard treatment in severe COVID-19 cases, and is considered to increase plasma levels of cystatin C. However, this has not been studied in COVID-19. Glomerular filtration rate (GFR) is a clinically important indicator of renal function, but often estimated using equations (eGFR) based on filtered metabolites. This study focuses on sources of bias for eGFRs (mL/min) using a creatinine-based equation (eGFR_LMR_) and a cystatin C-based equation (eGFR_CAPA_) in intensive-care-treated patients with COVID-19. This study was performed on 351 patients aged 18 years old or above with severe COVID-19 infections, admitted to the intensive care unit (ICU) in Uppsala University Hospital, a tertiary care hospital in Uppsala, Sweden, between 14 March 2020 and 10 March 2021. Dexamethasone treatment (6 mg for up to 10 days) was introduced 22 June 2020 (n = 232). Values are presented as medians (IQR). eGFR_CAPA_ in dexamethasone-treated patients was 69 (37), and 74 (46) in patients not given dexamethasone (*p* = 0.01). eGFR_LMR_ was not affected by dexamethasone. eGFR_LMR_ in females was 94 (20), and 75 (38) in males (*p* = 0.00001). Age and maximal CRP correlated negatively to eGFR_CAPA_ and eGFR_LMR_, whereas both eGFR equations correlated positively to BMI. In ICU patients with COVID-19, dexamethasone treatment was associated with reduced eGFR_CAPA_. This finding may be explained by corticosteroid-induced increases in plasma cystatin C. This observation is important from a clinical perspective since adequate interpretation of laboratory results is crucial.

## 1. Introduction

The coronavirus 2 (SARS-CoV-2) has spread over the world and caused nearly half a billion confirmed cases of COVID-19, out of which more than 6 million people are estimated to have died [1]. The SARS-CoV-2 infection is primarily a respiratory infection, but secondary effects can injure other organs, including the kidneys. Acute kidney injury (AKI) has considerable impact on the consequences of this viral infection, especially in critically ill patients. AKI induced by COVID-19 is believed to be caused by the interaction of several mechanisms, including dehydration causing impaired renal blood flow [2,3], coagulation activation with microthrombotization [3,4,5], and immune activation of inflammatory neutrophil polymorphonuclear cells [6,7]. Direct renal infection by SARS-CoV-2 has also been proposed but not verified [8]. In intensive-care-treated patients with COVID-19, the overall AKI incidence was initially nearly 90% [2]. There was a relationship between more severe AKI and worse outcome, even after adjusting for demographics, comorbid conditions, and illness severity [9].

Monitoring renal function is crucial in severe COVID-19 cases. Ideally, accurate determination of kidney function should easily be performed at low cost. A drawback of conventional analyses of glomerular filtration rate (GFR) is that they use endogenous analytes to calculate an estimated glomerular filtration rate (eGFR). Most equations for eGFR are based on the creatinine content in plasma, as a measure of how well waste products are filtered from the blood. However, numerous factors influence the plasma concentrations of creatinine, such as muscle mass, gender, age, and fluid and nutritional status. Furthermore, methods for creatinine calibration have improved and thereby have become more specific, which may lead to overestimation of GFR when some creatinine-based equations are used [10].

Cystatin C is produced by nucleated cells, freely filtered across the glomerular membrane, and thereby acting as an endogenous marker of GFR [11,12]. In patients with COVID-19, high levels of cystatin C independently predict the risk of developing a more serious disease and adverse outcome [13], which is in agreement with findings in unselected intensive care unit (ICU) patients, where the cystatin C-based equation predicted mortality better than a creatinine-based equation [14]. A shortcoming of cystatin C is the fact that this cysteine protease inhibitor may be influenced by corticosteroid treatment [15,16,17,18]. In 2020, dexamethasone was identified as being able to improve outcomes in hospitalized COVID-19 patients needing supplemental oxygen [19,20]. This implies a potential for underestimating eGFR when using the cystatine C CAPA (Caucasian, Asian, Pediatric, and Adult) equation [21] in the COVID-19 cohort. The effect of dexamethasone on eGFR_CAPA_ in COVID-19 has, to the best of our knowledge, not previously been evaluated.

The purpose of this study is to evaluate common sources of bias when determining eGFR, using a cystatin C-based equation [22] and a creatinine-based equation (the revised Lund–Malmö GFR estimating equation; LMR) [23] in critically ill COVID-19 patients, with special regard to the potential effects of dexamethasone.

## 2. Materials and Methods

### 2.1. Study Population

Patients aged 18 years old or above with severe COVID-19 infections, verified with a positive polymerase chain reaction (PCR) test of a nasopharyngeal sample, admitted to the intensive care unit (ICU) at Uppsala University Hospital, a tertiary care hospital in Uppsala, Sweden, between 14 March 2020 and 10 March 2021 were considered for inclusion in this study. Pregnancy was an exclusion criterion. In total, 370 patients were evaluated for inclusion in this prospective non-interventional study, which is a part of the Uppsala PRONMED-study cohort. Blood sampling for analysis of plasma creatinine and cystatin C was part of routine care of the patients. Median time with COVID-19 before ICU admission was 10 days (IQR:4). Demography within the material has previously been described [24,25].

From 22 June 2020 and onwards, all patients with COVID-19 that were admitted to ICU and needed supplemental oxygen therapy, received dexamethasone at 6 mg per day for up to 10 days.

In a very limited number of patients with exceptionally high BMI, higher doses of dexamethasone were given at the discretion of the attending physician.

### 2.2. Ethical Approval

This study was performed in accordance with ethical principles that have their origin in the Declaration of Helsinki [26] and consistent with ICH/GCP E6 (R2). The study was approved by the National Ethical Review Agency Dnr 2017-043 (with amendments 2019-00169, 2020-01623, 2020-02719, 2020-05730, 2021-01469) and 2022-00526-01.

Informed consent was obtained from all participating patients or given by proxy if the patient was unable to give consent. The protocol of the study was registered a priori at (Clinical Trials ID: NCT04316884). The study was performed according to relevant directives. The STROBE guidelines were followed in reporting [27].

### 2.3. Laboratory Analyses and Clinical Variables

Samples were taken at admission to the ICU as part of the routine procedure. Plasma creatinine (µmol/L) was analyzed at the department of clinical chemistry and pharmacology, Uppsala University Hospital, Uppsala, using an IDMS-calibrated enzymatic method on a Roche Cobas Pro (Roche Diagnostics, Rotkreuz, Switzerland). The laboratory is accredited by Swedac (Borås, Sweden). ISO 15189:2012 specifies the requirement for quality and competence in the laboratory, which is participating in Equalis (Uppsala, Sweden) external quality assurance programs for creatinine. Cystatin C was analyzed on an Architect ci16200 (Abbot Laboratories, Abbott Park, IL, USA) with IDMS-calibrated enzymatic creatinine reagents from the same manufacturer, and cystatin C reagents from Gentian AS (Moss, Norway). Cystatin C-based eGFR was calculated from plasma cystatin C by means of the International Federation of Clinical Chemistry equation CAPA (cystatin C-based eGFR_CAPA_) [28]. The applied equations [28,29] are shown in Table 1.

When the CAPA equation was developed, all cystatin C analyses were performed at the same laboratory as the present analyses [28]. Estimations of absolute GFR in mL/min were estimated using the CAPA equation [22] and the 2011 LMR equation [23]. Both equations primarily estimate relative GFR in mL/min/1.73 m^2^, and were thus deindexed for body surface area using the DuBois equation [30,31]. The CAPA equation had a median bias of −5.7 mL/min in the validation set of Swedish adults when the equation was developed [28]. In the combined Swedish CAPA development and validation cohort (n = 3495), median mGFR was 56 mL/min, whereas the median bias for LMR was 0.7 mL/min [32]. AKI stages were determined for each patient during ICU stay and related to severity as previously defined and classified [33]. A cystatin value of 1.64 mg/L (1.17–2.07) was set as a cutoff for AKI according to a study by Yildirim and co-workers in COVID-19 patients [34]. The logarithmic ratio of LRM/CAPA was used to relate any bias between the two eGFR equations, which may provide “standardization” across skewed data [35,36,37].

### 2.4. Statistical Analysis

Descriptive statistics are presented as the median and interquartile range (IQR) for continuous variables. Student’s *t*-test was used to compare the results of the two equations. Correlations between the equations were calculated using Pearson’s correlation. The Mann–Whitney U test was used to calculate the probability of chance differences in eGFRs in patients related to the sources of bias, including treatment with dexamethasone, gender, age, BMI, and maximal CRP within the cohort. Statistics were calculated using R (https://www.r-project.org, version 4.0.2, accessed on 29 August 2022). The value *p* < 0.05 was considered significant.

## 3. Results

In total, 351 patients aged 19–86 years (median age was 64 years and interquartile range (IQR) was (55-73 years) were included, 252 of which were males. Of the first 119 patients that did not receive dexamethasone, 22% were women. The corresponding ratio for those treated with dexamethasone was 31%. This difference was not significant. Age profiles (years) did not differ between males and females, regardless of treatment with dexamethasone (males 65 (57–72); females 63 (53–74)) or without (males (61 (51–71); females 66 (53–74)). Twenty-one percent of the patients in our cohort had cystatin C above 1.64 mg/L at ICU admission, and 60% had higher cystatin C values at any time during the entire ICU stay. In the whole cohort, eGFR_CAPA_ was 14% lower in patients treated with dexamethasone than in dexamethasone-naïve patients. The death rate among women was 14% and among men was 17%, across the whole study (n.s.). Similarly, maximal CRP was 212 (140–253) in women and 224 (129–306) in men (n.s.).

The median number of days with COVID-19 before admission to ICU was 10 days (IQR: 8–12). Median BMI (weight (kg) × height (m)^−2^) was 29 (IQR: 26–34). Thirty-four patients were treated with renal replacement therapy (median: 10 days; IQR: 3–15 days), and 173 patients were treated with artificial ventilation (median: 8 days; IQR: 5–15 days). Baseline data for included patients given dexamethasone (n = 232) or not (n = 119) are presented in Table 2. There were no significant differences in age or BMI between patients treated with dexamethasone or not. In contrast, the maximal CRP value was lower (*p* < 0.00001) in dexamethasone-treated patients than in patients not given any corticosteroid. Median BMI in 30-day survivors was 30 (IQR: 26–31). The corresponding median BMI of those who did not survive 30 days was 27 (IQR: 23–31) (*p* = 0.00008).

Figure 1 shows that GFR estimated by CAPA was lower in patients treated with dexamethasone than in those not given dexamethasone (*p* = 0.01). There was no difference in eGFR_LMR_ between dexamethasone-treated patients and those who did not receive this corticosteroid. Absolute eGFR_LMR_ was lower in males, which was in contrast to eGFR_CAPA_, where there was no difference between males and females. Age, BMI, and maximal CRP were associated with differences in both eGFR_LMR_ and eGFR_CAPA_. The logarithmized LMR/CAPA ratios show the magnitude of dexamethasone treatment, gender, age, BMI, and CRP when comparing these eGFRs.

Table 3 shows the population distribution with respect to eGFR, corticosteroid treatment, and gender. Gender had an impact on eGFR_LMR_, which was higher in females, both in dexamethasone-treated patients (*p* < 0.00001) and in patients not treated with corticosteroids (*p* = 0.0002). eGFR_CAPA_ was lower (*p* = 0.02) in men treated with dexamethasone (=173) than in dexamethasone-naïve men (n = 79). In dexamethasone-treated women (n = 77), eGFR_CAPA_ was lower compared to dexamethasone-naïve women (n = 22). This difference did not reach statistical significance. Gender did not significantly influence eGFR_CAPA_ in either dexamethasone-treated patients nor in dexamethasone-naïve patients.

## 4. Discussion

The main finding of the present study is that the sources of bias differ between LMR and CAPA in COVID-19 patients, and especially that the consistent use of dexamethasone in severe COVID-19 cases will tend to make cystatin C-based eGFR lower than in creatinine-based equations, and likely lower than the true GFR. This is in agreement with previous studies on the effects of corticosteroid treatment [15,16]. Zhu and co-workers [17] found, in an experimental study, that dexamethasone increases cystatin C levels in plasma, an effect suggested to be glucocorticoid-receptor-mediated, without any decrease in GFR, as evaluated by renal inulin clearance. Since dexamethasone caused significant and dose-dependent expression of cystatin C secretion in a cell culture [18], biotransformation does not seem to be of major importance for the effect of dexamethasone on cystatin C secretion. Gender had a significant impact on eGFR_LMR_, but not on eGFR_CAPA_, in our cohort.

AKI is a condition linked to increased severity of COVID-19, and increased mortality [38]. Elevated levels of creatinine are associated with impaired renal performance, increased risk of long-term dialysis dependence, increased severity of disease, and ultimately death [39]. In our cohort, the vast majority of ICU-treated patients with COVID-19 were men. This is in agreement with a study of Jin and co-workers [40], who found that females had better prognosis in COVID-19 than men irrespective of age. The more favorable outcome in women with severe COVID-19 may, at least partly, be explained by the homologous pair of X chromosomes inherited from both parents (maternal, Xm; paternal, Xp). The potentially protective protein thymosin beta-4 is, in humans, encoded by the TMSB4X gene and found on the X chromosome, where it escapes X-inactivation and could thereby help to explain a genetic advantage in COVID-19 [41]. Administration of thymosin beta-4 increases fibrinolysis [41]. In this context, it is interesting that the permeability-increasing effect of some plasmin-derived vasoactive peptides is augmented by an inhibitor of the angiotensin-converting enzyme [42].

Since many genes that express proteins involved in apoptosis and inflammation are located on the X chromosome, it is not unlikely that the observed differences in eGFR_LMR_ between men and women are due to an innate difference in the expression of these genes, and subsequently, variations of sexual hormones, which regulate different immunological functions. Oberholzer and co-workers [43] found that after severe sepsis, the incidence of multiple organ dysfunction syndrome, e.g., elevations in procalcitonin and interleukin-6, were significantly more frequent in males than in females. It is commonly assumed that this advantage is due to the effects of sex hormones, since half of the cells from females express either Xm or Xp, and therefore, females are cellular mosaics for their X-linked polymorphic genes, representing an advantageous innate immune response [44]. Hence, the phenotypic expression of X-linked alleles in males may be responsible for a less adaptive and balanced response to inflammatory conditions. Furthermore, hormonal differences may also be accountable for the manifestation of enzymatic proteins, e.g., nitric oxide synthase and superoxide dismutase [45]. Although the female gender seems to have a positive impact on myocardial inflammatory response to ischemia–reperfusion injury, which seems clinically relevant since postischemic myocardial function was significantly improved in females compared with males [46]; however, this does not seem to the case in sepsis [47]. In this context, it is interesting that, in a multicenter ICU database study, women had significantly lower creatinine levels, which did not affect the primary outcome variable, namely, ICU mortality [47], a finding that may be in accordance with the fact that increased plasma levels of creatinine do not necessarily reflect lower kidney function [48]. Additionally, a study in healthy humans revealed that proximal tubular tissue architecture adapts itself in accordance with the female reproductive hormone cycle [49]. Considering the age profile in our cohort this does not seem to be plausible explanation for higher eGFR_LMR_ in women. It is known that muscle mass affects the creatinine levels [50,51]. In healthy young adults, lean tissue mass correlates to creatinine, but not to GFR, when determined by creatinine [52]. In patients, even with a substantially decreased GFR, creatinine levels may still be within the normal range [53]. Hence, eGFR_LMR_ seems to represent a more complex pattern, involving extrarenal factors indirectly related to GFR (e.g., age, weight, muscle mass, gender, atherosclerosis, endothelial dysfunction, inflammatory and hemostatic events), thereby limiting the usefulness of this endogenous biomarker in severe diseases.

We did not notice any effect of gender on eGFR_CAPA_, which is in agreement with a previous study by Grubb et al. [28]. However, Hanneman et al. [54] found, in a population-based cohort study, that creatinine-based equations (Cockcroft–Gault and MDRD) of eGFR were higher in men, while eGFR based on the cystatin C formula was higher in women [55]. Pottel et al. [56] found, in a large general population study including renal disease patients, that mean (SD) measured GFR (mL/min/1.73 m^2^) in females older than 70 years was 57.7 (20.3) and 53.7 (21.0) in men. These differences in results may, at least partly, be explained by the eGFR_LMR_ equation, since male gender and elevated creatinine may indicate the progression of COVID-19 [57]. Hence, the difference in eGFR_LMR_ between men and women in our cohort, may be explained by a more severe progression of COVID-19 in men.

Furthermore, there were marked negative correlations between age and eGFR_LMR_ and eGFR_CAPA_, respectively. These negative correlations are in agreement with previous findings in primary care patients [58].

Both absolute eGFR_LMR_ and eGFR_CAPA_ correlated positively to BMI in our cohort. Although obesity has a negative impact on renal function [59,60], we noted absolute eGFR_LMR/CAPA_ values, which even in the BMI range of 50–60 were higher than the average absolute eGFRs. In this context, it should be noted that visceral adipose tissue predicts disease risk and mortality better than BMI [61], and that the assessment of compartmental adipose tissue in patients hospitalized with COVID-19 turned out to be independent prognostic factors [62]. Low relative eGFR predicts worse outcomes in COVID-19 [63] cases, but it is a drawback of relative eGFR that these equations usually are body surface area (BSA) indexed, causing a risk of underestimation of GFR in the obese and the contrary in those with underweight [64]. Thus, it seems reasonable to assume that absolute eGFR should be preferred, not only when dosing and evaluating toxicity of renally excreted drugs, but also in reducing biases due to divergencies in BSA [32]. This may be supported by the fact that we found a 10% higher BMI in those surviving 30 days compared to those who did not, which also might argue in favor using deindexed equations.

Maximal CRP during ICU stay correlated inversely to both eGFR equations. This may reflect an effect of a downgraded inflammatory response after administration of dexamethasone, an assumption supported by the fact that maximal CRP was significantly lower in dexamethasone-treated ICU patients with COVID-19. A confounding factor when evaluating the effect of CRP on eGFRs, is that there was a variating timespan from administration of dexamethasone until analysis of the maximal CRP value.

The kidneys have undergone evolutionary progresses enabling most mammals to exist on land where water and salts must be retained, while waste is concentrated and excreted, a process where fluidal chemical composition as well as osmotic pressure are of uttermost importance. Considering these circumstances it is not surprising that differences in eGFR are related to the equations in question. From a clinical perspective it is important to determine biases between various equations in the subtle interplay, where GFRs are estimated. Dosage of several, potentially toxic drugs, are based on renal function, which therefore should be assessed as adequately as possible.

Altogether, we advocate the use of cystatin C in favor of creatinine if the cystatin C assay is available with similar test turnaround times as for creatinine. This cysteine protease inhibitor is expressed by nucleated cells, being a sensitive and reliable marker of GFR [11].

Cystatine C has a predictive value in COVID-19 [13,34,65,66] and also in ICU patients in general [14,21,67]. Cystatin C has the drawback of being affected by corticosteroids [15,16,17], but not by CRP [68]. Creatinine is frequently used to determine renal function, but shortcomings are interferences by gender, age, nutrition, and muscle mass [51,69]. When creatinine is determined by isotope dilution mass spectrometry (IDMS) calibration, the creatinine values were reduced by approximately 25%, but with wide variations. However, the Cockcroft–Gault equation is still widely used, which may lead to an overestimation of renal performance [10]. From an economical perspective, the cost of cystatin C is comparable to that of enzymatically determined creatinine [68].

This study has some limitations. It is a single-center study, where data were collected during a period where therapy was improved and experience and knowledge of managing COVID-19 was increased. Although the data were collected prospectively, this analysis was performed retrospectively. Since the number of patients that fell ill with COVID-19 varied during this period, ICU admission might have been biased. The ratio of included women was higher in those treated with dexamethasone than in dexamethasone-naïve women. Although this difference did not reach statistical significance, it could hypothetically be a confounding factor.

## 5. Conclusions

In ICU patients with COVID-19, treatment with dexamethasone reduced eGFR_CAPA_ compared to eGFR_LMR_, which should be taken into consideration when renally excreted, and potentially toxic, drugs are used. Although this study was performed for severely ill COVID-19 patients, this postulate is unlikely to be limited to such patients. Female gender was associated with increased eGFR_LMR_. BMI correlated positively with both eGFR equations, whereas age and maximal CRP during ICU stay correlated negatively. These findings may have important impacts on the interpretation of laboratory analytes as well as clinical decision making.

## Figures and Tables

**Figure 1 biomedicines-10-02708-f001:**
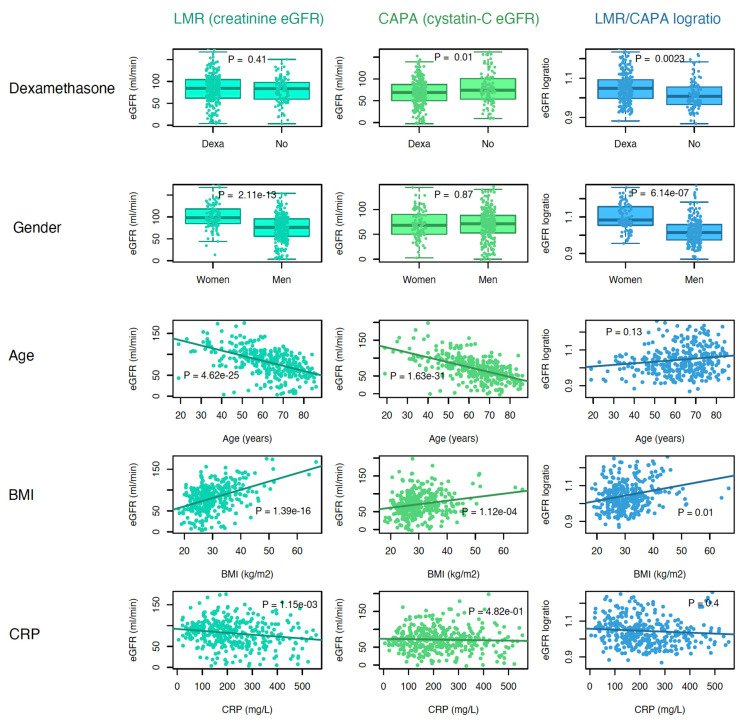
eGFR_CAPA_ was lower in dexamethasone-treated patients, whereas eGFR_LMR_ was lower in men. Age (years) and maximal CRP (mg × L^–1^) were associated with declining eGFR irrespective of equation, whereas increasing body mass index (BMI; weight (kg) × height (m) ^–2^) correlated to higher eGFRs.

**Table 1 biomedicines-10-02708-t001:** Equations used to estimate glomerular filtration rate (eGFR) by cystatin C and creatinine. Both equations depend on age, whereas LMR also depends on gender. pCr is the plasma creatinine level.

	Equations	
eGFR_CAPA_	130 × plasma Cystatin C^−1.069^ × Age^−0.117^ − 7	
eGFR_LMR_	e^X^ − 0.0158 × Age + 0.438 × ln(Age)	
		Variables
	X = 2.50 + 0.0121 × (150 − pCr)	Female pCr < 150
	X = 2.50 − 0.926 × ln(pCr/150)	Female pCr > 150
	X = 2.56 + 0.00968 × (180 − pCr)	Male pCr < 180
	X = 2.56 − 0.926 × ln(pCr/180)	Male pCr > 180

**Table 2 biomedicines-10-02708-t002:** Relationships between absolute GFR in mL/min, estimated by the CAPA equation [22] and the 2011 LMR equation [23], age (years), body mass index (BMI; weight (kg) × height (m)^−2^, and CRP maximal value (mg × L^−1^) during ICU stay.

	No Steroids		Dexamethasone	
	Median	IQR	Median	IQR
eGFR_LMR_	84	39	85	42
eGFR_CAPA_	74 *	46	69	37
Age	61	22	65	18
BMI	29	8	29	8
CRPmax	280 **	169	181	143

* Denotes *p* = 0.01 between dexamethasone-naïve and dexamethasone-treated patients. ** Denotes *p* < 0.00001 between dexamethasone-naïve and dexamethasone-treated patients. Otherwise, no significant differences were noted.

**Table 3 biomedicines-10-02708-t003:** Pattern of ICU-treated COVID-19 patients based on absolute eGFR (mL/min), with respect to gender and dexamethasone treatment; females vs. males.

	Females		Males	
	eGFR_LMR_ ^§^		eGFR_LMR_	
Dexamethasone	Median	IQR	Median	IQR
	100	38	76	40
	eGFR_CAPA_		eGFR_CAPA_ ^$^	
	Median	IQR	Median	IQR
	67	40	69	34
	eGFR_LMR_ ^#^		eGFR_LMR_	
No Steroids	Median	IQR	Median	IQR
	94	20	75	38
	eGFR_CAPA_		eGFR_CAPA_	
	Median	IQR	Median	IQR
	70	34	76	48

**^§^** Denotes *p* < 0.00001 vs. males; **^#^** Denotes *p* = 0.0002 vs. males. **^$^** Denotes *p* = 0.02 vs. eGFR_CAPA_ in men.

## Data Availability

Datasets used and/or analyzed during the current study are available from the corresponding author on reasonable request (https://doi.org/10.17044/scilifelab.14229410 accessed on 18 March 2021).

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
