# Peer review of "Differential Bias for Creatinine- and Cystatin C- Derived Estimated Glomerular Filtration Rate in Critical COVID-19"

_biomedicines, 2022, doi:10.3390/biomedicines10112708_

Round 1

Reviewer 1 Report

In this work, the authors have compared the estimation of glomerular filtration using creatinine and Cystatin-C. They show significant differences between the results obtained with Cystatin in patients treated with dexamethasone, with lower levels of eGFR in patients treated with Dexa; this is all the more marked in women. This could be due to an increase in serum cystatin induced by dexamethasone treatment. 

Major comments:

- The degrees of renal insufficiency are characterized by different thresholds. Instead of simply considering the median values, it could be interesting to indicate how many patients would have a higher degree of renal insufficiency by considering the values ​​obtained with Cystatin C.

- The reviewer is not convinced by the conclusion that female gender was associated with increased eGFRlmr. It could be more relevant to observe that in this population, females have a normal EGFRlmr, in contrast to men. Is it statistically true , It not indicated in the table 3. if the difference is statistically significant, it should be precized in the table 3; But as it is written in the discussion, the lower level of eGFRlmr could be due to the fact that men have usually more severe forms of COVID; Therefore, the description of the population, in terms of gender, should be detailed: how many died ? how was the CRP as welle as the CRP max in males and females ?

- Tables and their legend  should be improved (for exemple, the significance of CRPmax in not indicated in table 2), similar rearks for table 3 concerning the statistical significances,.

- in the introduction, page 2, line 70, it is written that the specific effect of dexamethasone on Cystastine C has not been evaluated, whereas in the discussion (p7, line 211), it is written, using the same references, that dexamethasone increase cystatin C secretion.

- At least, which test is the more relevant , creatinine or Cystatin C ? 

- Minor comments:

Abbreviations are missing : p2, line 45 "NLRP, line 70 "CAPA" or repeated "ICU" is indicated line 65, p2, and repeated line 81, p2

Reviewer 2 Report

This is a very interesting issue in the field of Covid-19 disease and Acute Renal  Injury however I have major concerns and comments on this study.

Comments

1. The length of time patients received dexamethasone prior to calculation of e-GFR is not reported, it was just one dose or a course of therapy?

2. It is common knowledge that e- GFR is affected by gender, however just age is reported. What was the male to female ratio ? This could be a confounding factor to be considered.

3. The LRM/CAPA model was used for this study. Is there any reference? Please add the reference or justify the use of this formula. Please clarify it  in methods section.

4.Table 2 : You should add/report the p-value

5. Table 3 : You should add/report the p-value

6. Regarding the analysis of e-GFR there are  some serious omissions. Does the e-gFR based on  two methods differ in each subgroup? A comparison should be made in all groups and in addition in each subgroup according to gender, age, dexamethasone intake. Υou should add them in order to complete the analysis.

7. Conclusion paragraph must me improved . The authors should state the usefulness of their work. How can the results of this work be utilized in clinical practice?

General Comment

The study was really interesting but both  the methodology and  statistical analysis need major revisions in order to obtain results with clinical value.

Round 2

Reviewer 2 Report

I have no further comments